# *Swap Up* Your Meal: A Mass Media Nutrition Education Campaign for Oklahoma Teens

**DOI:** 10.3390/ijerph191610110

**Published:** 2022-08-16

**Authors:** Dana E. Wagner, Gabrielle Seneres, Elisabeth Jones, Kelli A. Brodersen, Sjonna Whitsitt-Paulson

**Affiliations:** 1Rescue Agency, PBC, 2437 Morena Blvd, San Diego, CA 92110, USA; 2Oklahoma Tobacco Settlement Endowment Trust, 2800 N. Lincoln, Ste 202, Oklahoma City, OK 73105, USA

**Keywords:** obesity prevention, nutrition, nutrition education, mass media public health, public health campaigns, digital and social media public health campaigns

## Abstract

To address a statewide need for obesity prevention, the Oklahoma Tobacco Settlement Endowment Trust launched *Swap Up* in 2021, a mass media nutrition education effort for teens, ages 13–18. *Swap Up* utilizes the SAVI messaging approach, an audience-centric message development framework that recognizes barriers to healthy living and offers realistic solutions. Five months into the campaign, an online survey was conducted (*n* = 200) to assess short-term program goals related to campaign delivery, engagement, and relevance. A secondary, long-term goal related to documenting and understanding self-reported changes in past month nutrition-related behaviors was also explored. A majority of participants (72%) reported aided awareness of the campaign brand logo/advertisements, and awareness (83%) of at least one main message. Nearly half (44%) of the participants reported at least one engagement with digital media. Main message recognition, perceived relevance, and self-reported nutritional behaviors were consistently highest among those reporting both campaign awareness and digital engagement. Ultimately, *Swap Up* reached and delivered nutrition education messages to Oklahoma teens within the first year of launch, as intended, and was associated with self-reported changes in recent behavior. This study provides evidence that SAVI offers a promising approach for nutrition education, and underscores why digital and social media engagement strategies are critical for mass media teen behavior change campaigns. Campaign implementation and evaluation are ongoing.

## 1. Introduction

Oklahoma is among the states affected by disproportionately high rates of adult obesity, ranking fourth highest in the nation in 2021 [1]. Obesity poses major risks to both long- and short-term health outcomes, such as heart disease, high blood pressure, depression, eating disorders, and 13 different types of cancer [2,3,4]. Obesity among youth is a concern due to its likelihood to continue through adulthood [5]. States across the U.S. continue to see high rates of obesity in children and adults, including Oklahoma [6,7].

For Oklahomans, in particular, there are structural and systemic barriers that make healthy living an uphill battle. Oklahoma is one of many southern, largely rural states that have a high density of food deserts, meaning there is restricted access to healthy and fresh foods [8]. Oklahoma’s poverty rate is consistently higher than the national average [9] and studies show that children in homes of lower socioeconomic status (SES) are more likely to be overweight or obese [10]. This is largely because high-calorie foods and drinks are more affordable than healthier alternatives [11,12]. High caloric intake is also exacerbated by the availability of fast food in the state; Oklahoma is ranked the fourth highest state for number of fast food restaurants (5.3 restaurants per 10,000 people) [13,14]. Issues of food insecurity were also amplified by the economic impact of the COVID-19 pandemic. In August 2020, approximately one in eight Oklahomans reported sometimes or often not having enough food [9,15].

While these challenges are contributing to high obesity rates in the state, it is important to consider the extent to which public health can help adolescents overcome barriers and be empowered to make healthier nutritional choices in their day-to-day lives. Research shows that establishing healthy nutrition and physical activity habits earlier in life can reduce the risk of adulthood obesity regardless of current weight status [16,17]. Therefore, providing adolescents with education and an easily adoptable framework for making daily nutritious choices could have benefits beyond their teen years.

### 1.1. Mass Media Obesity Prevention

Obesity prevention and nutrition education in Oklahoma requires a large-scale, awareness-driving approach that will reach all teens, regardless of current weight or living environment. Historically, most successful obesity prevention efforts originated in schools or community-based settings, educating younger children or families about nutrition or implementing physical activity programs in small groups [18,19,20,21,22,23]. Government organizations and commercial marketers also teamed up to promote healthy nutritional intake through mass media, such as milk (“Got Milk?”) and fruit/veggies (“Five A Day”) [24]. However, many of these campaigns were designed to promote product sales and their outcomes for public health and obesity prevention are not well documented [25]. One notable exception is the CDC VERB campaign, which was a multi-media social marketing campaign aimed at increasing physical activity among tweens (ages 9–13) [26,27]. The campaign showed year-over-year successes with audience activity levels increasing alongside increases in campaign awareness [26]. To our knowledge, a similar mass media approach was never applied to public health nutrition education, though there is evidence of its promise [28,29].

Digital media, especially social media, provides an increasing number of unique opportunities to reach teens through everyday media consumption behaviors [30,31,32]. While the measurement of digital media delivery and consumption is lagging behind [33,34,35], more research is emerging that demonstrates the ability of technology to successfully deliver persuasive health messages to large populations of young people [36,37]. Evans and colleagues describe an ecological approach to obesity prevention that includes multi-channel, mass-marketing strategies to promote the modeling of healthy behaviors and audience engagement. An ecological approach considers both the social and physical environments and challenges surrounding individuals as primary determinants of health [28,38]. In a state like Oklahoma where obesity prevention is a statewide, population-level need, it stands to reason that an ecological mass-media approach to educational messaging could help adolescents to better navigate their daily nutrition.

### 1.2. Fast Food and Sugary Drink Marketing

It is important to note that the successful messaging about food and beverages to young people by commercial industries spans decades. Studies show that fast food companies, especially McDonald’s, use child-targeted marketing strategies (“Happy Meals”) worldwide with price promotions disproportionately targeting low-income communities, which further supports higher risk of obesity for lower SES children [39]. Television advertisements of unhealthy foods are also associated with increased consumption of calories, which are often lacking in nutrients and are insufficient food sources for young people [40,41]. In fact, Nickelodeon was shown to promote largely unhealthy foods, such as sugar-sweetened beverages and candy or fruit snacks, with 65% of their food ads featuring items of poor nutritional quality and limited ads for fruits or vegetables [42]. In 2018, beverage companies spent USD 20.7 million advertising children’s drinks with added sugars, and sugary drinks made up 62% of the USD 2.2 billion in total U.S. children’s drink sales [43]. Therefore, it is not surprising that a 2010 study found 27% of children’s daily calories come from snacking, with sugar-sweetened beverages and desserts as the highest sources of caloric intake [44].

The mass media and audience-centric strategies employed by the commercial marketing industry further exacerbate challenges for obesity prevention. This lifelong exposure to food and beverage marketing heavily influenced nutritional beliefs and habits in today’s youth, whereby exposure to healthy food marketing is an exception. Still, commercial practices could offer insights into effective marketing and counter-marketing strategies.

### 1.3. The SAVI Framework

Rescue Agency (Rescue) developed the SAVI messaging approach to break down complex behaviors and help identify new, simple solutions towards the goal of healthier outcomes. SAVI identifies four areas of strategic messaging, which include: specific, acceptable, viable, and impactful (see Table 1). SAVI assumes that lifestyle change is too daunting or unrealistic for an audience’s circumstances or current ability and helps identify appropriate educational messages that address their daily reality.

SAVI draws on existing theories of behavior change commonly used in prevention and intervention efforts, such as Social Cognitive Theory and Social Norm Theory. Social Cognitive Theory states that human behavior is the result of the dynamic interaction between personal (thoughts and feelings), behavioral (health knowledge/skill) and environmental factors (external) [45,46]. Self-efficacy, or one’s confidence in their ability to take action on the specific behavior and, the perceived benefits of taking the action are critical to behavior change. SAVI addresses these factors by ensuring health messages provide a clear and logical plan of action for changing behavior, which are tailored to the perceived benefits, personal motivations, and challenges of the audience [47]. Social Norm Theory posits that incorrect perceptions of similar others’ behaviors normalize and justify an individual’s own unhealthy behavior [48,49]. Research shows that perceptions about normative behavior among peers is related to sugary drink and unhealthy snack consumption, as well as body mass index (BMI) [49,50,51]. Similarly, SAVI assumes that people are emotional and socially driven decision makers who need to see themselves and their circumstances reflected to care about a message. Therefore, it is important to conduct audience research to uncover the deeply held beliefs, norms, motivators, and perceived barriers surrounding current nutrition, and potential pathways to behavior change.

### 1.4. Oklahoma’s Swap Up Campaign for Teens

Acknowledging both the current health status of Oklahomans and the common nutritional challenges they face, a state agency, the Tobacco Settlement Endowment Trust (TSET), launched a statewide effort to prevent and reduce obesity among adolescents (ages 13–18) in Oklahoma. *Swap Up* leans on principles of SAVI to offer Oklahoma teens relevant and realistic tips related to daily nutritional choices (see Table 1). The campaign seeks to reduce and prevent obesity by encouraging incremental lifestyle changes, beginning in teen years, a time when young people gain increased autonomy and choices regarding meals outside of the home. *Swap Up* is unique in two major ways. The first is its practical and audience-forward approach to nutrition for teens. The second is that it is the first known public health-driven obesity prevention effort that attempts to reach an entire state with mass media messaging, encompassing both urban and rural audiences.

The long-term goal of the campaign is to reverse obesity trends by increasing consumption of fresh produce and water and decreasing sugary drink consumption. The campaign set short-term goals related to campaign and message awareness, relevance of the messages, and documenting audience engagement with the campaign. Throughout the campaign, self-reported behaviors are tracked in order to see trends in health-related behaviors over time.

### 1.5. Current Study

In the first year of the campaign, the program sought to investigate whether the *Swap Up* campaign successfully reached its short-term goals related to successful media delivery (50% campaign and message awareness), perceived relevance by the audience, and evidence that the audience was engaging with campaign assets. As a secondary research question related to long-term goals, we investigated whether Oklahoma teens considered or tried to increase healthy nutritional behaviors and decrease unhealthy nutritional behaviors. We expected that those aware and/or engaged with the campaign to be most likely to report considering or trying to engage in healthy behaviors, compared to those unaware or unengaged.

## 2. Materials and Methods

### 2.1. Campaign Intervention

*Swap Up* (SwapUpOK.com, accessed on 31 May 2022) was developed for Oklahoma TSET by Rescue and launched in February 2021. *Swap Up* was created to complement a current, statewide, adult-focused effort called *Shape Your Future*, which educates parents and adults on nutrition inside and outside of the home. Following principles of SAVI, *Swap Up* development included formative research to better understand the motivations and barriers underlying Oklahoma teens’ nutritional choices.

In August–September 2020, a cross-sectional online survey was conducted (*n* = 403) with Oklahoma teens to assess the status of their nutrition-related beliefs and behaviors, as well as future intentions to change their behavior [52]. Importantly, when teens reported they did not eat vegetables or fruits in the past 7 days, it was most commonly because they did not think about it (43% for vegetables and 53% for fruits) and their family did not buy it (38% for vegetables and 38% for fruits). Additionally, 78% of overweight teens reported that there are fruits and vegetables available where they buy their food, but only 60% reported that there are fruits and vegetables available in their home. Rural teens were significantly less likely than urban teens to report that vegetables are available where they buy food (71% vs. 82%), that their family can afford to buy fruits and vegetables (63% vs. 79%), and that there is a large selection of fruits and vegetables where they live (60% vs. 72%; all *p*s < 0.05). Teens overall and overweight teens, in particular, reported high levels of intent to change their behaviors in the next 7 days, with 89% reporting they would drink more water, 62% reporting they would eat more vegetables, and 63% reporting they would eat more fruits. These data provided context for the everyday challenges teens face and provided hope that Oklahoma teens do desire better nutritional outcomes.

As a follow-up, a series of in-depth interviews were conducted with Oklahoma teens (*n* = 21) in October–December 2020 to test draft campaign messages [53]. The sample was drawn from both urban (*n* = 12) and rural counties (*n* = 9), and 70% of participants reported being overweight. Teens were most receptive to messages about the short-term benefits and immediate consequences of healthy nutritional choices. They confirmed that they know and understand the long-term consequences of unhealthy nutrition, such as obesity and risk of various diseases, but were most persuaded by messages that convinced them of how their immediate nutritional choices can impact them *today*. Following this, the campaign messages were designed to focus on how nutrition-rich foods/drinks help them think, act, and feel good, while nutrition-poor foods make them think, act, and feel bad. Teens connected to messages portraying everyday activities that teens care about and relate to, such as sports and school performance. Additionally, teens enjoyed light-hearted messages that demonstrated facts with humor, so they did not feel as though the message was telling them what to do. As expected, teens also expressed little perceived control over their nutritional choices at home. Even when they eat out with friends and expressed that fast food and convenience stores were readily available, accessible, easy hang out spots, and within their budget. Messages that acknowledged the conveniences provided by these venues while providing tips on swapping side dishes or drinks with healthy options, instead of skipping fast food altogether, were thought to be helpful, feasible, and relevant.

Given the high risk of adulthood obesity among teens in the state and similar perceptions of draft campaign messages across obese and non-obese teens in formative research, the campaign was designed to reach Oklahoma teens of varying levels of obesity risk with a single message. However, teens’ perceptions of message context, tone, and setting did differ between urban and rural counties. Source characteristics, such as the narrator’s accent and style were most important to legitimize the nutrition message for rural teens. That is, rural teens were most receptive to messages delivered by teens who looked and sounded like they were from rural areas. For urban teens, the setting was important for receptivity. They wanted to see teens in places that were realistic, such as a fast-food restaurant or a school. As a result, a body of campaign advertisements was developed to maximize relevance to rural versus urban audiences, with targeted media delivery to appropriate counties.

*Swap Up* is primarily a digital and social media campaign directed towards teens 13–18 years old that includes active Instagram, YouTube, Facebook, and Snapchat accounts, and an interactive website. The social posts are designed to either drive traffic to the YouTube page or website and/or to promote engagement with short videos, memes, or GIFs that deliver facts. Across assets for all three message platforms that were launched in Year 1 of the campaign (Fuel for Football, Sluggish, and Blank-Minded; see Table 2), there were a total of 43 million video impressions and 40,000 website visits. Rescue and TSET worked with an Oklahoma media agency, VI Marketing and Branding, to deliver *Swap Up* advertisements on broadcast television, Over The Top (OTT) streaming services, and radio on stations and programs consumed by parents and influential adults. Given cultural and systemic barriers to healthy nutrition, the goal is to ensure visibility and support for *Swap Up* among the broader Oklahoma population.

### 2.2. Design and Procedures

In June–July 2021, approximately 5 months after the launch of *Swap Up*, a cross-sectional survey was administered (*n* = 200) to assess evidence of campaign and message awareness, relevance, and engagement. Additionally, early evidence of long-term goals was assessed to track nutrition-related behaviors among the audience. The majority of participants (74%, *n* = 148) were recruited via online advertisements on Facebook and Instagram. Recruitment advertisements directing individuals to the screening survey were served with age and geo-targeting features. Respondents answered screening questions to ensure they were age 13 to 18 years and Oklahoma residents.

An additional 26% (*n* = 52) of the sample was retained from the pre-launch survey in 2020. Following survey completion, participants who were interested were directed to a separate survey where they could sign up for future studies. Age qualification criteria were adjusted to allow those who were 18 years old in the initial survey, and were 19 years old at the first follow-up survey, to participate. All participants received a USD 15 online gift card. The study protocol was approved by the Advarra Institutional Review Board. Participants received a study information sheet and provided informed assent or consent, depending on their age.

### 2.3. Measures

*Demographics.* Participants reported demographics including age, sex, race/ethnicity, and zip code of residence. Participants were labeled as “Urban” or “Rural” based on population density of the corresponding county [54]. See Figure 1 for county designations.

*Obesity Risk.* Adapted from the Oklahoma Youth Risk Behavioral Survey (YRBS; [7]), participants were asked to describe their weight on a five-point scale (5—very overweight, 4—slightly overweight, 3—about the right weight, 2—slightly underweight, and 1—very underweight). Responses were dichotomized as Overweight (4–5) or Not Overweight (1–3).

*Swap Up Main Message Awareness.* Prior to brand or advertisement questions, participants were asked, “Have you seen any ads with the following messages or taglines?” without revealing the source. Ten phrases featured in *Swap Up* advertisements were presented with a prompt to select all that apply (See Appendix A for items). Two types of variables were calculated: (1) dichotomized *Swap Up* Main Message Aware; and (2) summed score (0–10) indicating number of *Swap Up* main messages recognized.

*Increasing Water.* Participants who reported considering or trying to increase the “amount of water” they drank in the past 30 days (from YRBS [7]), were provided with a multi-select list of 11 items assessing which of the following actions they took (see Appendix A for items). A summed score was created, ranging from 0 to 11.

*Decreasing Sugary Drinks.* Participants who reported considering or trying to decrease the “amount of soda/pop or other sugary drinks (e.g., Coke, milkshake, Frappuccino, slushies)” they consumed over the past 30 days (from YRBS [7]), were provided with a multi-select list of 14 items assessing which of the following actions they took (see Appendix A for items). A summed score was created, ranging from 0 to 14.

*Increasing Fruits and Veggies.* Participants who reported considering or trying to increase the “amount of fruits/vegetables” they ate over the past 30 days (from YRBS [7]), were provided a multi-select list of 12 items assessing which of the following actions they took (see Appendix A for items). A summed score was created, ranging from 0 to 12.

*Decreasing Greasy, Fried, and Sugary Foods.* Although tangentially related to campaign goals, we sought to document reports of decreasing unhealthy food consumption. Participants who reported considering or trying to decrease the “amount of greasy, fried or sugary foods” they ate over the past 30 days (from YRBS [7]), were provided with a multi-select list of 14 items assessing which of the following actions they took (see Appendix A for items). A summed score was created, ranging from 0 to 14.

*Swap Up Aided Campaign Awareness.* To assess brand awareness, participants were presented with a series of 8 health and nutrition-related brand logos and asked if they heard of each brand, with “Yes,” “No,” and “I don’t know” as response options. To assess ad awareness, participants were shown two of three campaign advertisements, which included “Fuel for Football” to all participants, and either “Sluggish” for Urban teens or “Blank Minded” for Rural teens. After each ad, participants were asked “Have you seen this video before?” with “Yes,” “No,” and “I’m not sure” as response options. Participants who selected “Yes” to the aided brand or aided video ad awareness questions were designated as *Swap Up* Campaign Aware.

*Swap Up Digital Engagement.* Next, participants were asked, “Have you ever seen, interacted, or shared any *Swap Up* social media posts, videos, or advertisements?” Participants were provided a list of 5 methods of digital interaction or engagement with a prompt to select all that apply (See Appendix A for items). A dichotomous variable was created where any selected response was considered *Swap Up* Engaged.

*Swap Up* Awareness/Engagement. To help address moderate multi-collinearity and increase sample sizes for subgroups, a 3-level analytic variable was created that combined campaign awareness (Y/N) and digital engagement (Y/N). Based on responses for the two items, participants were categorized as Aware/Engaged (*n* = 75), Aware/Not Engaged (*n* = 67), or Not aware (combination of not aware/engaged and not aware/not engaged; *n* = 57).

*Swap Up Perceived Relevance to Nutritional Mindset.* After viewing two *Swap Up* video ads, participants were asked to rate agreement on a five-point scale (“Strongly Agree” to “Strongly Disagree”) for 3 items assessing *Swap Up*’s relevance to them personally. Statements included “*Swap Up* feels like it’s for people like me,” “*Swap Up* offers a new way to look at nutrition,” and “*Swap Up* has information that could be helpful in improving my nutrition.” A composite variable was created by averaging the 3 items (Chronbach’s *ɑ* = 0.75).

### 2.4. Data Analysis

Frequencies were generated in SPSS for demographics and Year 1 campaign outcomes related to media delivery, relevance, and engagement. Statistical comparisons were conducted between campaign aware/not aware, engaged/not engaged, urban/rural county, and overweight/not overweight on message awareness, perceived relevance, and nutritional behaviors to understand patterns (chi-squared tests of independence for dichotomous outcomes and t-tests for continuous outcomes). To test whether awareness of main messages, perceptions of campaign relevance, and considering/trying to increase nutritional behaviors differed based on campaign awareness/engagement, we conducted a series of MANCOVAs, controlling for obesity risk and county.

## 3. Results

### 3.1. Demographics

Two hundred participants ages 13–19, living in Oklahoma, were recruited. The mean age of the sample was 16.8 (*SD* = 1.4), with a majority being female (70%) and White/Caucasian (59%). The final sample included 73% urban (*n* = 145) and 27% rural (*n* = 55) participants, with 44% reporting to be overweight (see Table 3). Demographic distributions of the sample were similar across urban and rural counties. However, non-Hispanic Black teens were more likely to live in an urban county than a rural county (9% vs. 2%, *p* < 0.05). The race and ethnicity composition of the sample was reflective of the state of Oklahoma in 2021 more broadly (e.g., 63.8% White/Caucasian, 7.8% Black/African American, 11.7% Hispanic, etc.) [55].

### 3.2. Campaign Delivery, Engagement, and Relevance

A majority of participants (72%, *n* = 143) reported aided awareness of the *Swap Up* brand and/or video advertisements, and 83% reported awareness of at least one campaign main message. The average number of messages recognized was 3.9 (*SD* = 2.8) and the most recognized was “Water refreshes you” (46%) followed by “Fruits and veggies give you energy” (45%). See Appendix A for main message awareness frequencies by item and subgroup. Of note, 5 out of 10 items significantly differed based on campaign awareness and 8 out of 10 items significantly differed based on digital engagement.

On average, teens rated *Swap Up*’s perceived relevance to their nutritional mindset at 3.8, (*SD* = 0.8). Nearly half of the participants (44%) reported at least one *Swap Up* digital engagement (*M* = 0.7, *SD* = 0.9). The most commonly endorsed items were “Seen a GIF/short video from *Swap Up* online” (22%) and “Opened or seen a story on social media from *Swap Up*” (22%). See Appendix A for digital engagement frequencies by item and subgroup. Of note, two out of five items significantly differed based on campaign awareness and all five items significantly differed based on overall digital engagement status.

### 3.3. Nutrition-Related Behaviors

*Increasing Water.* Eighty-three percent of participants reported trying or considering at least one water consumption-increasing behavior in the past 30 days (*M* = 2.4, *SD* = 2.2). The most commonly endorsed items were “Set a goal to drink more water” (62%), followed by “Purchased, or asked others to purchase for me sparkling unsweetened water or bottled water” (35%). One out of eleven items significantly differed based on campaign awareness and five out of eleven items significantly differed based on digital engagement. See Appendix A for frequencies by item and subgroup.

*Decreasing Sugary Drinks.* Sixty percent of participants reported trying or considering at least one sugary drink-decreasing behavior in the past 30 days (*M* = 2.2, *SD* = 2.6). The most commonly endorsed items were “Set a limit on how many sodas/sugary drinks I should drink each day, or limited the times/days I drink them” (45%), followed by “While I still had these drinks, I tried to drink a bit less of them by taking a smaller serving or not finishing all of the drink” (36%). Two out of fourteen items significantly differed based on digital engagement. See Appendix A for frequencies by item and subgroup.

*Increasing Fruits and Veggies.* Sixty-seven percent of participants reported trying or considering at least one fruit/vegetable consumption-increasing behavior in the past 30 days (*M* = 2.7, *SD* = 2.7). The most commonly endorsed items were “Set a goal to eat more fruits/vegetables” (44%), followed by “Told my family about my plans to eat more fruits/vegetables” (35%) and “Looked up recipes or information about how to prepare fruits/vegetables” (35%). Three out of twelve items significantly differed based on digital engagement. See Appendix A for frequencies by item and subgroup.

*Decreasing Greasy, Fried, and Sugary Foods.* Sixty-five percent of participants reported trying or considering at least one unhealthy food-decreasing behavior in the past 30 days (*M* = 2.7, *SD* = 2.9). The most commonly endorsed items were “While I still had these foods, I tried to eat a bit less of them by taking a smaller serving or not finishing all of the food” (53%), followed by “Set a goal to eat less of these foods” (48%). Four out of 14 items significantly differed based on digital engagement. See Appendix A for frequencies by item and subgroup.

### 3.4. Do Message Awareness and Perceived Relevance Differ by Campaign Awareness and Engagement?

To better understand the role of campaign exposure and digital engagement on awareness and relevance outcomes, the combined *Swap Up* awareness/engagement variable was entered into a MANCOVA with main message awareness and perceived relevance to nutritional mindset as dependent variables, controlling for obesity risk and county; Pillai’s Trace = 0.17, *F*(390) = 9/15, *p* < 0.001. Significant univariate main effects of awareness/engagement were detected for both dependent variables (*p*s < 0.05). Pairwise comparisons revealed that (a) both aware/engaged and aware/not engaged teens recognized significantly more unaided main messages compared to not aware, and (b) aware/engaged teens rated campaign relevance significantly higher than not aware (see Figure 2 for all comparisons).

### 3.5. Do Self-Reported Nutrition-Related Behaviors Differ by Campaign Awareness and Engagement?

Finally, to better understand the role of campaign exposure and digital engagement on nutrition-related behaviors, the combined *Swap Up* awareness/engagement variable was entered into a MANCOVA with four dependent variables (increasing water, decreasing sugary drinks, increasing fruits and veggies, and decreasing greasy, fried, and sugary foods) and two covariates (obesity risk and county); Pillai’s Trace = 0.114, *F*(386) = 2.92, *p* < 0.01. Significant univariate main effects of awareness/engagement were detected for all four dependent variables (*p*s < 0.05). Pairwise comparisons revealed that aware/engaged teens were more likely to report trying or considering increasing water and decreasing sugary beverages in the past 30 days compared to teens who were aware/not engaged and not aware. Aware/engaged teens were also more likely to report trying or considering increasing fruits and veggies in the past 30 days compared to teens not aware. Finally, aware/engaged teens were significantly more likely than aware/not engaged and marginally (*p* = 0.08) more likely than not aware teens to report trying or considering decreasing greasy, fried, and sugary foods in the past 30 days (see Figure 3 for all comparisons).

## 4. Discussion

### 4.1. Summary of Findings

The current study found evidence that *Swap Up* successfully reached and delivered nutrition education messages to Oklahoma teens within its first five months of implementation. As a comparison point, the CDC’s VERB campaign reached the same level of aided awareness (72%) after two years [26]. Overall, teens felt the campaign was relevant and made for people like them. This finding is especially hopeful for an audience that has a series of major barriers in the way of behavior change and is resistant to being “told” what to do.

Importantly, we found that various campaign outcomes and recent changes in self-reported behaviors/intentions were most pronounced among participants both aware and engaged with the campaign compared to those not aware. From a campaign planning perspective, this suggests that engagement-driving media approaches could increase receptivity and eventual behavior change. Similar to principles of the Elaboration Likelihood Model, campaign planners should consider the extent to which engaging social media assets and interactive games that utilize key messages may provide valuable opportunities for education and persuasion [56]. As *Swap Up* implementation continues, it will be critical to continue measuring self-reported digital and social media engagements across campaign assets in order to better understand the relationship between engagement and behavior change over time.

The ultimate goal of *Swap Up* is to see incremental trends of sustained, self-reported behavior change, leading to prevention of adulthood obesity. In order to support this goal, *Swap Up* is regularly refreshing content across brand platforms. Since the launch of the first advertisements (Fuel for Football, Sluggish, and Blank-Minded), a new series of video ads were introduced every six months. Additionally, new social and digital content is regularly posted in order to mimic other brands and influencers that teens care about, aligning with how they expect to consume media. This strategy is intended to increase the likelihood of grabbing teens’ attention and creating opportunities for multiple engagements because there is consistently new content to interact with. With each new impression or engagement, *Swap Up* is reinforcing relevant information, increasing the likelihood of multiple exposures/engagements, and most importantly, normalizing nutritional messaging and healthy swaps in teens’ everyday lives [57,58]. The hope is that campaign media and messaging will reinforce teens’ current intentions to make healthy choices, ultimately leading to behavior change [57,58,59].

It is possible that the observed effects of campaign engagement on teens’ behaviors are actually in the inverse direction; teens who engaged with *Swap Up* may do so because they are already receptive to nutritional messages and to increasing nutritional behaviors prior to exposure. In this case, one would expect that teens who are not overweight may be more aware/receptive to the campaign or be more likely to report increasing or considering nutritional behaviors than overweight teens. However, we did not observe systematic differences by self-reported obesity risk in the current sample, and instead utilized this variable as a covariate to increase the accuracy of the statistical models. It will be important to continue tracking campaign outcomes year over year, as well as individual characteristics of teens, to better understand and observe the potential impact of *Swap Up* on Oklahoma teens.

### 4.2. SAVI Messaging

This is the first campaign to design, implement, and measure outcomes based on SAVI messaging, an approach developed by Rescue Agency [47]. The empathetic approach provided by SAVI led to insights and messaging that directly addressed teens’ everyday nutritional challenges. This study provides promising evidence that SAVI can successfully be applied to nutrition education, that SAVI messages are memorable, and that an intended audience is receptive to this approach. More research on SAVI is needed to better understand its impact on campaign outcomes.

### 4.3. Limitations and Future Research

As with all studies, there are limitations to note. The cross-sectional design and convenience sampling of the current research limit generalizability and attribution of outcomes to the campaign or underlying messaging framework. Future studies utilizing a longitudinal design would allow for a more concrete understanding of behavior change mechanisms at play. Additionally, the reliance on self-reported behavioral data may introduce error through recall bias or social desirability. Future research should consider the extent to which real-time reporting of actual daily nutritional intake would more accurately measure teen behavior change as a result of campaign exposure. Lastly, the campaign was implemented and the study was conducted during the COVID-19 pandemic, which may have temporal effects on nutrition-related behaviors and/or media consumption [60,61]. Thus, findings would potentially be different if this campaign and/or study was implemented a year earlier or later. The campaign will continue to collect data annually in order to assess mid-term and long-term outcomes.

### 4.4. Conclusions

*Swap Up* is the first known mass media nutrition education and obesity prevention campaign to attempt to reach adolescents across an entire state, including rural and urban audiences. The current study found evidence that, within five months of implementation, *Swap Up* successfully reached and educated Oklahoma teens about nutrition, and exceeded its Year 1 goals related to media delivery and receptivity. More importantly for the future of Oklahomans’ health, *Swap Up* awareness and online campaign engagement was associated with an elevated number of self-reported healthy behaviors compared to those not aware. These findings provide a promising foundation for the campaign’s long-term goals of documenting incremental changes in teens’ behavior and obesity prevention. Ultimately, this research underscores the importance of the purposeful design and measurement of digital and social media engagement strategies within teen behavior change campaigns.

## Figures and Tables

**Figure 1 ijerph-19-10110-f001:**
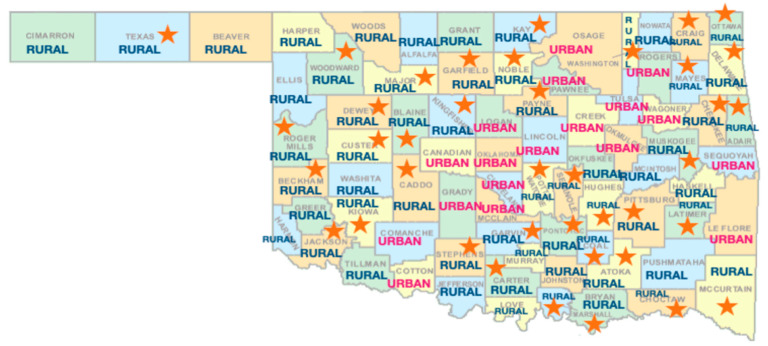
Map of county designation. Note: Star indicates at least 1 participant self-reported a zip code from that county within the study.

**Figure 2 ijerph-19-10110-f002:**
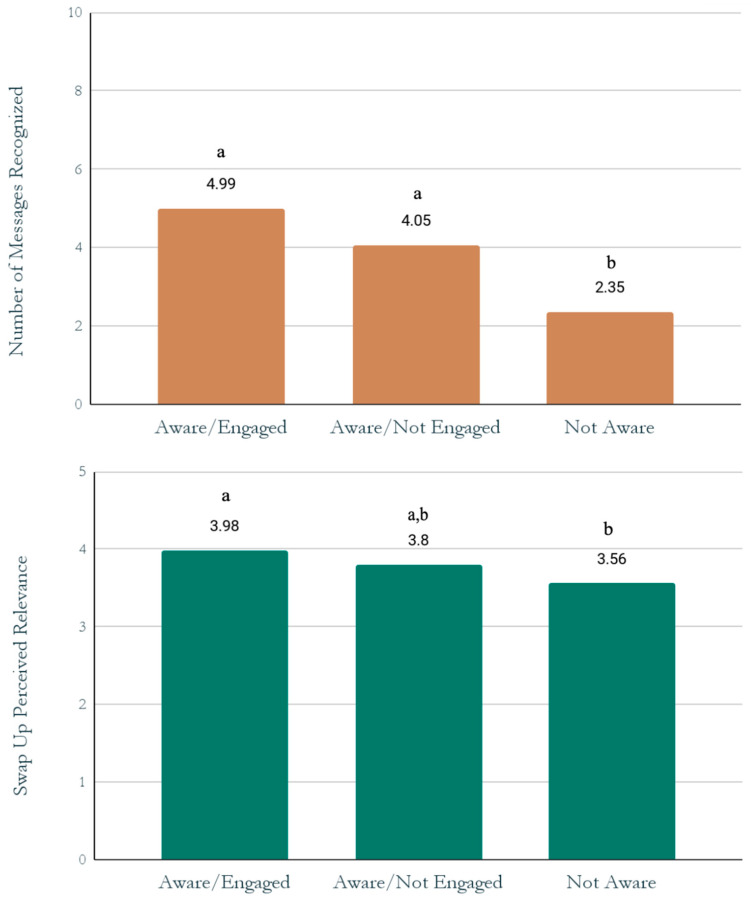
*Swap Up* main message awareness and perceived relevance by campaign awareness/engagement, controlling for obesity risk and county. Note: columns with different letters are significantly different (*p* < 0.05); columns with the same letter do not differ significantly from each other.

**Figure 3 ijerph-19-10110-f003:**
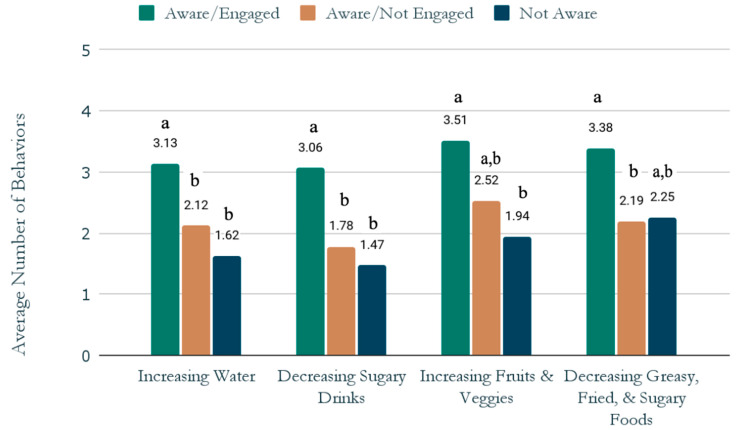
*Swap Up* awareness/engagement on nutrition-related behaviors controlling for obesity risk and county. Note: for each behavior, columns with different letters are significantly different (*p* < 0.05); columns with the same letter do not differ significantly from each other.

**Table 1 ijerph-19-10110-t001:** Applying SAVI to *Swap Up*.

SAVI Component	Description	*Swap Up* Application
Specific	Must include a specific example of what the audience can do to change their behavior	When you go to the gas station for breakfast, get a low-fat and low-sugar yogurt and a banana instead of a donut.
Acceptable	Must be acceptable within the cultural, familial, and social contexts of the audience’s lives	Swap fizzy water for fountain soda at a fast food restaurant (instead of “Don’t eat at fast food restaurants or convenience stores”).
Viable	Must be realistic within the constraints of our audience’s available time, budget and skills	Do not expect a teen to be able to revamp their family meals or overhaul their diet and instead focus on a reasonable individual action they can control, like what they buy outside of the home and what is within their budget.
Impactful	If adopted, the message would cause a meaningful impact on the audience’s nutrition	Pick a side of carrots instead of fries (this is impactful because fast food is a daily occurrence).

**Table 2 ijerph-19-10110-t002:** Examples of *Swap Up* Year 1 campaign assets.

Asset	Media Type	Audience	Main Message	Media Flighting
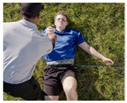 Fuel for Football https://www.youtube.com/watch?v=99Ks8sG4dng accessed on 31 May 2022	:30 video ad	Statewide	Swap a slushie for water; sugar slows you down and causes crashes/water energizes you	February–March 2021
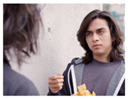 Sluggish https://www.youtube.com/watch?v=1u-pjQI9NyQhttps://youtu.be/ONDIIPbUxKI accessed on 31 May 2022	:30 video ad	Urban	Swap chips for apples and peanut butter; greasy foods slow you down/protein gives you energy	May–June 2021
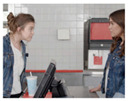 Blank-Minded https://www.youtube.com/watch?v=IlgYwdg0DCg accessed on 31 May 2022	:30 video ad	Rural	Swap a milkshake for water; sugar slows you down/water refreshes you	May–June 2021
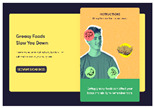 How Food Affects You	Interactive website/ game	Statewide	Greasy foods slow you down; sugar causes crashes	February 2021–current
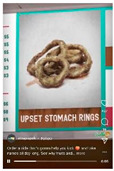 Dynamic Duo https://www.instagram.com/p/CU8raPQNnxf/?utm_medium=copy_link accessed on 31 May 2022	:15 digital video	Statewide	Fruits and vegetables give you energy; swap fried foods for a salad	October–December 2021
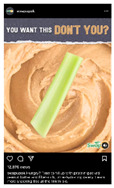 Snacking Tips https://www.instagram.com/swapupok/ accessed on 31 May 2022	GIF	Statewide	Nutrient-packed snacks will fill you up	October–December 2021

**Table 3 ijerph-19-10110-t003:** Sample Characteristics.

Variable	Total Sample % (*n*)
Total	200
Female	70% (140)
Age	16.8 (*SD* = 1.4)
13 years old	2% (4)
14 years old	6% (11)
15 years old	9% (18)
16 years old	22% (44)
17 years old	24% (48)
18 years old	33% (66)
19 years old	4% * (9)
Race/ethnicity	
Hispanic	11% (21)
Non-Hispanic White	59% (118)
Non-Hispanic Black	7% (14)
Non-Hispanic Asian/Pacific Islander	2% (7)
Non-Hispanic American Indian/Native American	9% (17)
Non-Hispanic other or 2 or more races	12% (23)
County (Urban)	71% (142)
Obesity Risk (Overweight)	44% (87)

* Participants who were 18 years old at the pre-launch survey were invited to participate in the follow-up.

## Data Availability

The data presented in this study are available on request from the corresponding author.

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
