# Peer review of "Swap Up* Your Meal: A Mass Media Nutrition Education Campaign for Oklahoma Teens"

_ijerph, 2022, doi:10.3390/ijerph191610110_

Round 1
Reviewer 1 Report
The manuscript entitled “Swap Up Your Meal: A Mass Media Nutrition Education Campaign for Oklahoma Teens” addressed the issue of higher obesity rates in teens in Oklahoma state. The Oklahoma Tobacco Settlement Endowment Trust launched the ‘Swap Up’ campaign for teens ages 13-18 in 2021, that utilizes SAVI messaging approach. To evaluate the implementation of this campaign an online survey of 200 individuals was conducted to assess campaign delivery, engagement, and relevance. And self-reported changes in nutritional-related behaviors was explored. Authors have found that within five months of implementation, ‘Swap Up’ successfully reached and educated Oklahoma teens about nutrition.
1. The language of the manuscript is clear and understandable.
2. The abstract represents the accurate summary of the research and results with proper explanation.
3. The authors have explained the limitation of this study in discussion.
In my opinion, the manuscript is suitable for publication in ‘International Journal of Environmental Research and Public Health’ journal.
Reviewer 2 Report
This is a very well-written and needed campaign. I have a few thoughts.
Table 8: I would make this supplemental. It's a lot of data to take in and with it's current location, breaks up the narrative.
Demographics of sample: Suggest adding sentence in results that indicates how representative this is of the general OK sample of teens. Respondents are primarily White/Caucasian--did the campaign speak more strongly to that demographic or is that demographic reflective of the majority of OK teens?
In Discussion: Add thoughts on how to improve behavioral change component. The data reports on "more likely to report trying or considering." For those who are "considering" they have yet engaged in desired behavior. Any thoughts on how to help teens make this leap?
Reviewer 3 Report
Please check the file attached herewith.
